# A Mixture of Polyunsaturated Fatty Acids ω-3 and ω-6 Reduces Melanoma Growth by Inhibiting Inflammatory Mediators in the Murine Tumor Microenvironment

**DOI:** 10.3390/ijms20153765

**Published:** 2019-08-01

**Authors:** Ewin B. Almeida, Karina P.H. Silva, Vitoria Paixão, Jônatas B. do Amaral, Marcelo Rossi, Roberta A. Xavier-Navarro, Karina V. Barros, Vera L.F. Silveira, Rodolfo P. Vieira, Luis V.F. Oliveira, Elizabeth C. Perez, Miriam G. Jasiulionis, André L.L. Bachi

**Affiliations:** 1ENT Research Lab., Department of Otorhinolaryngology -Head and Neck Surgery Federal University of São Paulo (UNIFESP), São Paulo 04039-002, Brazil; 2Samaritan Hospital Blood Bank, São Paulo 01232-910, Brazil; 3Healthcare Nutrition Science Manager, Danone Nutricia, São Paulo 01311-000, Brazil; 4Department of Biological Science, Federal University of São Paulo (UNIFESP), São Paulo 09972-270, Brazil; 5Brazilian Institute of Teaching and Research in Pulmonary and Exercise Immunology (IBEPIPE), São Paulo 12245-520, Brazil; 6Post-Graduation Program in Bioengineering, Universidade Brasil, São Paulo 08230-030, Brazil; 7Post-Graduation Program in Sciences of Human Movement and Rehabilitation, Federal University of São Paulo (UNIFESP), São Paulo 11015-020, Brazil; 8School of Medicine, Anhembi Morumbi University, São Paulo 12230-002, Brazil; 9University Center of Anápolis-UniEvangélica, Anápolis 75083-515, Brazil; 10Environmental and Experimental Pathology, Paulista University (UNIP), São Paulo 04057-000, Brazil; 11Laboratory of Ontogeny and Epigenetics, Pharmacology Department, Federal University of São Paulo (UNIFESP), São Paulo 04044-020, Brazil

**Keywords:** cyclooxygenase (COX), lipoxygenase (LOX), acute inflammation, leukotrienes, prostaglandins, cytokines, melanoma

## Abstract

Background: Although it has been previously demonstrated that acute inflammation can promote the tumor growth of a sub-tumorigenic dose of melanoma cells through of 5-lipoxygenase inflammatory pathway and its product leukotriene B_4,_ and also that the peritumoral treatment with eicosapentaenoic acid and its product, leukotriene B_5_, reduces the tumor development, the effect of the treatment by gavage with omega-3 and omega-6 in the tumor microenvironment favorable to melanoma growth associated with acute inflammation has never been studied. Methods: C57BL/6 mice were coinjected with 1 × 10^6^ apoptotic cells plus 1 × 10^3^ viable melanoma cells into the subcutaneous tissue and treated by gavage with omega-3-rich fish oil or omega-6-rich soybean oil or a mixture of these oils (1:1 ratio) during five consecutive days. Results: The treatment by gavage with a mixture of fish and soybean oils (1:1 ratio) both reduced the melanoma growth and the levels of leukotriene B_4_ (LTB_4_), prostaglandin E_2_ (PGE_2_), PGE_2_/prostaglandin E_3_ (PGE_3_) ratio, and CXC ligand 1 (CXCL1) and increased the levels of interleukin 10 (IL-10) to IL-10/CXCL1 ratio in the melanoma microenvironment. Conclusion: The oral administration of a 1:1 mixture of fish oil and soybean oil was able to alter the release of inflammatory mediators that are essential for a microenvironment favorable to the melanoma growth in mice, whereas fish oil or soybean oil alone was ineffective.

## 1. Introduction

Since Virchow described the presence of an intense leukocyte infiltration in tumors, the contribution of inflammatory factors to tumorigenesis and tumor progression became one of the main aspects studied in cancer. It is widely accepted that several chronic inflammatory diseases, such as gastritis, esophagitis, prostatitis, and chronic skin ulcers, among others, are associated with tumor development [1].

Inflammation is a biological response to tissue repair, in which there is not only the activation of different types of local cells, but also migration of leukocytes from the vascular system to the lesion sites. When associated with cancer, chronic inflammatory responses can contribute to tumor development and progression to create favorable conditions. In the tumor microenvironment, the inflammatory mediators released by normal or tumor cells are involved in the recruitment and activation of different types of leukocytes, which release both pro- and anti-inflammatory cytokines in this site. For instance, it has been demonstrated that neutrophils can be recruited from blood to tumor microenvironments by a variety of chemokines, such as CXC ligand 8 (CXCL8) or interleukin 8 (IL-8), growth-regulated oncogene alpha (GRO-α), and CXCL1, among others, and these cells can favor the tumor progression by releasing different types of cytokines, such as tumor necrosis factor alpha (TNF-α), interleukins (ILs) -1beta (1β), 6, 15, 17, and 18 [2].

Despite several studies that demonstrated how chronic inflammation contributes to the transformation, proliferation, and progression of many tumors, the effect of acute inflammation in tumor development is still poorly understood. Regarding this fact, we have demonstrated that the acute inflammatory response is able to create favorable conditions to the tumor growth of a sub-tumorigenic dose of murine melanoma cells [3]. In addition, our findings showed that this acute inflammatory response support to the tumor growth, dependent of the 5-lipoxygenase (5-LOX) pathway activation, was mainly mediated by leukotriene B_4_ (LTB_4_), whereas the activation of cyclooxygenase (COX) 1 or 2 was not relevant in this context [3].

It is known that the production of LTB_4_ from 5-LOX activity is dependent of the bioavailability of arachidonic acid, a molecule associated with a pro-inflammatory response, which is produced from linoleic acid metabolism, a polyunsaturated fatty acid classified as omega-6 [4,5]. On the other hand, eicosapentaenoic acid (EPA) and docosahexaenoic acid (DHA), two molecules derived from the alpha-linolenic acid metabolism, polyunsaturated fatty acid classified as omega-3, are also targets of 5-LOX activity, and the mediators produce anti-inflammatory actions [5]. Based on this information, Bachi et al. [3] showed that the peritumoral treatment with EPA, as well as with leukotriene B_5_ (LTB_5_), the main mediator derived from EPA after 5-LOX activity, was able to reduce tumor growth in our experimental model.

In order to gain insights into the effect of omega-3 and omega-6 in the growth of a sub-tumorigenic dose of melanoma cells dependent on an acute inflammatory response, in this study, mice were treated by gavage with omega-3-rich fish oil or omega-6-rich soybean oil, or a mixture of these oils in a ratio of 1:1, and the tumor growth, as well as the concentrations of inflammatory mediators (LTB_4_, LTB_5_, prostaglandin E_2_ (PGE_2_), and prostaglandin E_3_ (PGE_3_)) and cytokines (IL-6, IL-10, and CXCL1) in the tumor microenvironment, were evaluated.

## 2. Results

### 2.1. In Vitro Treatment with Omega-3 or Omega-6 Did Not Impair B16F10 Cells Viability

As shown in Figure 1, the in vitro treatment of B16F10 cells with different concentrations (40, 20, 10, and 5 μL) of fish oil rich in omega-3 (Figure 1A) or soybean oil rich in omega-6 (Figure 1B), or a mixture of these oils in a ratio of 1:1 (Figure 1C), for 48 h did not affect the viability of B16F10 cells. However, the treatment with higher concentrations of soybean oil and with the mixture induced tumor cell proliferation showed the same effect, whereas the fish oil treatment did not show the same effect.

### 2.2. In Vivo Treatment with a Mixture of Omega-3 Andomega-6 Reduced Tumor Growth

Based on the observation that the higher doses of fish oil rich in omega-3 or soybean oil rich in omega-6 or a mixture of these oil in a ratio of 1:1 did not affect the B16F10 cells viability, we initiated in vivo treatments by gavage of mice with a dose of 40 μL or 20 μL per day of oils. However, we observed that these doses induced severe diarrhea in the mice. Therefore, all the following in vivo results showed in this study were obtained using a dose of 10 μL per day of treatment. Figure 2 shows that only the treatment by gavage with the oil mixture was able to reduce the tumor growth.

### 2.3. Treatment with a Mixture of Fish and Soybean Oils Reduced the Pro-Inflammatory Mediators in the Tumor Microenvironment

After observing that the treatment by gavage with a mixture of fish oil rich in omega-3 and soybean oil rich in omega-6 in a ratio of 1:1 reduced the tumor growth, we performed a new round of in vivo experiments to evaluate the effects of the oil treatments in the tumor microenvironment favorable to the tumor growth and progression. A total of 21 days after the coinjection, the tumors from each mice group were removed in order to obtain a tumor homogenate that was used to determine the levels of inflammatory mediators LTB_4_, LTB_5_, PGE_2_, and PGE_3_ in the tumor microenvironment.

As shown in Figure 3, lower levels of LTB_4_ (*p* < 0.01, Figure 3A) and PGE_2_ (*p* < 0.001, Figure 3D) were found in the tumor homogenate from the mice group treated with a mixture of both oils in a ratio 1:1 as compared to control group. In relation to the levels of LTB_5_ (Figure 3B) and PGE_3_ (Figure 3E) found in the tumor homogenate, no differences were observed between the mice groups. Regarding the ratio between the levels of LTB_4_ and LTB_5_ (LTB_4_/LTB_5_, Figure 3C) in the tumor homogenate, no difference was found, but the ratio between the levels of PGE_2_ and PGE_3_ (PGE_2_/PGE_3_, Figure 3F) observed in the mice group treated with the mixture of fish and soybean oil (1:1 ratio) was lower than the control group (*p* < 0.01).

### 2.4. The Treatment with a Mixture of Fish and Soybean Oils Reduced the CXCL1 Levels in the Tumor Microenvironment

Figure 4 shows that CXCL1 levels found in the tumor homogenate from the mice group treated with a mixture of fish oil rich in omega-3 and soybean oil rich in omega-6 in a ratio of 1:1 were lower compared to control group (*p* < 0.01, Figure 4A). In relation to IL-6 levels (Figure 4B) found in the tumor homogenate, no differences were observed between the mice groups. The analysis of IL-10 levels (Figure 4C) showed that the mice groups treated with fish oil (*p* < 0.05) or soybean oil (*p* < 0.05) or the mixture of these oils (1:1 ratio, *p* < 0.01) showed increased levels of this anti-inflammatory cytokine in the tumor homogenate compared to control group. In addition, the ratio between the levels of IL-10 and CXCL1 (IL-10/CXCL1, Figure 4D) observed in the mice group treated with the mixture of fish and soybean oil (1:1 ratio) was higher than the control group (*p* < 0.05). No differences were observed in the ratio between IL-10 and IL-6 (IL-10/IL-6, Figure 4E).

### 2.5. In Vitro Treatment with Soybean Oil Associated or not with Fish Oil, but not with Fish Oil Alone, Induced IL-10 Release from B16F10 Cells

Based on the findings that the treatment by gavage with fish, soybean oil, or the oil mixture alters the cytokines levels in the melanoma microenvironment, we performed in vitro assays in order to evaluate if the treatment with these oils could induce a release of cytokines from B16F10 cells. As shown in Figure 5, higher IL-10 levels (Figure 5A) were found after the in vitro treatment of B16F10 cells with 20 and 10 μL of soybean oil or a mixture of the soybean oil and fish oil in a ratio of 1:1 for 48 h compared to the control. No differences were observed in the levels of IL-6 (Figure 5B) and CXCL1 (Figure 5C) in B16F10 cells after oil treatment.

## 3. Discussion

In this study, we showed that the treatment with a mixture of fish oil rich in omega-3 and soybean oil rich in omega-6 in a ratio of 1:1 during five consecutive days after the coinjection of B16F10 cells in the subcutaneous tissue of C57BL/6 mice was able to reduce not only the tumor growth, but also the levels of pro-inflammatory mediators (LTB_4_ and PGE_2_) and CXCL1 in the tumor microenvironment.

It is widely accepted that the chronic presence of infections and inflammation can favor the development of cancer [6]. Regarding human melanoma, its development is not necessarily associated with the presence of inflammatory processes. However, studies have highlighted the importance of an inflammatory microenvironment and some innate immunity components for the melanoma onset and progression. For instance, Culp et al. [7], through proteomic analysis, showed overexpression of inflammatory proteins from 3 to 7 days after the injection of B16F10 cells in C57BL/6 mice. Corroborating this observation, the same “inflammatory gene signature” found in the primary site of the melanoma cells injection was also identified in the sentinel lymph nodes that had metastatic cells, whereas in the lymph nodes without metastatic cells this inflammatory signature was not observed [8]. In addition, Correa et al. [9] showed that the presence of an acute inflammatory response is essential to create a favorable microenvironment to the growth of a sub-tumorigenic dose of melanoma cells.

Therefore, it is reasonable to suggest that the presence of anti-inflammatory agents in the tumor microenvironment of B16F10 cells could alter the favorable conditions to the melanoma growth. In order to confirm this suggestion, we were able to demonstrate that the treatment by gavage with a mixture of fish oil rich in omega-3 and soybean oil rich in omega-6 in a ratio of 1:1 was able to reduce the melanoma growth. Corroborating our results, among several studied nutritional strategies, balancing the tissue ratio between the omega-3 and omega-6 intake, for instance in 1:1 or 1:2 ratios and not exclusively with one type of these fatty acids, it has been able to suppress the development of many chronic diseases [10,11]. Although studies that aimed to evaluate the effect of omega-3 in melanoma development and progression are scarce, it was shown that there is a lower incidence of cancer in Asian populations due to a daily consumption of marine foods rich in omega-3 [12,13]. It has been proposed that the increased bioavailability of omega-3, and consequently their metabolites (EPA and DHA), could suppress carcinogenesis through the competition between these molecules and arachidonic acid by the COX and LOX enzymes. This could lead to a reduction in the pro-carcinogenic eicosanoids production, especially PGE_2_ [14], associated or not to an increase of anti-inflammatory molecules, such as LTB_5_ and/or PGE_3_. 

Based on the information cited above, we evaluated how the treatment with the mixture of omega-3 and omega-6 (1:1 ratio) alters the favorable conditions to the melanoma growth through of analysis of the inflammatory mediator (LTB_4_, LTB_5_, PGE_2_, and PGE_3_) and cytokines (IL-6, IL-10, and CXCL1) levels in the melanoma microenvironment. 

Our results showed a significant reduction of LTB_4_ and PGE_2_ levels in the tumor tissue homogenate from mice treated with the mixture of omega-3 and omega-6 (1:1 ratio) compared to the control group, whereas LTB_5_ and PGE_3_ levels were similar between the groups. In addition, the group treated with the mixture of omega-3 and omega-6 (1:1 ratio) showed a lower ratio between the levels of PGE_2_ and PGE_3_ (PGE_2_/PGE_3_) when compared to the control group. These data corroborate with the previously mentioned research and allow us to suggest that the increased bioavailability of omega-3 together with omega-6 in a ratio of 1:1 could increase the EPA and DHA levels, which could compete with arachidonic acid (AA) by LOX and COX enzymes and consequently lead to the reduction of the pro-inflammatory mediators, LTB_4_ and PGE_2_, as observed in the present study.

In a similar way, the treatment by gavage with the mixture with omega-3 and omega-6 (1:1 ratio) also altered the cytokines levels in the melanoma microenvironment, since the CXCL1 levels were significantly reduced in this group compared to the control group. 

It is widely known that during inflammatory responses, both innate and adaptive immune system cells have the capacity to secrete different types of cytokines that exhibit a prominent regulatory activity (positive or negative) when interacting with their ligands or receptors in target cells [15]. Regarding the chemokines, these molecules are a large cytokine family that demonstrate similar structure and function [16,17,18]. In general, these molecules are composed of low molecular weight polypeptides and act mainly by controlling the migration of leukocytes to the inflammation site [19]. These chemokines are characterized by the presence of two cysteine residues, which can be together (CC subfamily) or separated by another amino acid (CXC subfamily). In relation to the CXC subfamily, these molecules act preferentially as a neutrophils chemoattractant [20,21]. In addition, it was also demonstrated that CXC chemokines could also act as potent angiogenic factors [22]. In relation to this angiogenic property, it was reported that when CXCL1 produced by melanoma cells binds to CXCR1 or CXCR2 receptors expressed in endothelial cells, it can induce the recruitment of these cells to the tumor microenvironment, leading to the formation of new vessels [23,24]. Therefore, the lower CXCL1 levels found in the tumor microenvironment of the mice group treated with the mixture of the omega-3 and omega-6 in a ratio of 1:1 can putatively decrease not only the neutrophil recruitment but also the neo-angiogenesis process, which consequently could lead to the reduction of tumor growth, as demonstrated here. Considering that melanoma cells can secrete CXCL1 [25], it was mandatory to analyze not only whether the B16F10 cells used in this study could produce this chemokine, but also whether this production could be impaired by fatty acids treatment. After 48 h of fatty acids treatment, no differences in the in vitro CXCL1 levels were observed. This observation reinforces our suggestion that the treatment by gavage with a mixture of omega-3 and omega-6 (1:1 ratio) alters the tumor microenvironment favorable to melanoma growth and does not act directly in the B16F10 cells.

Besides the evaluation of CXCL1 levels in the melanoma microenvironment, we also analyzed the effect of the fatty acids treatment on in vitro and in vivo levels of IL-6, a pro-inflammatory cytokine [26], and IL-10, a classical anti-inflammatory cytokine [27].

In relation to the context of IL-6 and melanoma cancer, while increased expression of this cytokine in the melanoma microenvironment has been shown to favor the occurrence of tumor cachexia [28] and metastasis [29,30], it was reported that omega-3 and its metabolites can decrease tumor growth of both human non-melanoma skin carcinoma cells [31] and melanoma cells [32] through the reduction of pro-inflammatory cytokines levels, such as IL-6, in the tumor microenvironment. However, we did not observe alterations in the IL-6 levels of both in vitro and in vivo treatment with the fatty acids omega-3 or omega-6.

For IL-10 levels, in a different way for the described for IL-6 levels, the in vitro results showed that the addition of omega-6 or the oil mixture, but not omega-3 alone, in the B16F10 cell culture was able to promote the elevation of IL-10 levels. It has already been shown that B16F10 cells produce IL-10, which can act in an autocrine manner, stimulating its proliferation [33]. These findings allow us to suggest that the increased proliferation of B16F10 cells observed with the treatment with omega-6 and oil mixture were influenced by the elevation in the IL-10 levels. Interestingly, mice treated by gavage with fish oil rich in omega-3, soybean oil rich in omega-6, or the mixture between these oils with a ratio of 1:1 showed higher IL-10 levels in the tumor microenvironment compared to the control group. In this context, Tuccitto et al. [34] mentioned that, due to the extreme heterogeneity of the melanoma microenvironment, it is possible that the actions of IL-10 in this microenvironment may be subjected to the influence of other substances and may alter the plasticity modeling of the melanoma. Interestingly, it was demonstrated that omega-3 can increase IL-10 levels in skin cancer and its elevation was able to inhibit the pro-inflammatory cytokines production, leading to reduced tumor growth and progression [31,35]. Based on these observations, we analyzed the ratio between the levels of IL-10 and the other cytokines (IL-6 and CXCL1) in the melanoma microenvironment. A higher ratio was found only between IL-10 and CXCL1 levels (IL-10/CXCL1) in the group of animals treated with the mixture of fish oil rich in omega-3 and soybean oil rich in omega-6 (1:1 ratio) compared to the control mice group. This finding evidences that treatment with the mixture of the oils containing omega-3 and omega-6 increases IL-10 levels in a way that could negatively influence CXCL1 production in a melanoma microenvironment.

Figure 6 shows a representative illustration highlighting the main results found. In summary, the oral administration of the 1:1 mixture of fish oil and soybean oil was able to decrease melanoma growth in an experimental model, in which the acute inflammatory response was essential to create favorable conditions to the tumor growth.

## 4. Materials and Methods

The study was carried out after the approval of the Ethics Committee on Animal Use (CEUA) of the Cruzeiro do Sul University under the number 032/2016 on 7 December 2016. All experimental protocol was in agreement with the Brazilian guidelines for scientific animal care and use [36].

### 4.1. Mice, Reagents, Cell Culture, and Cell Viability Assay

The murine melanoma cell line, B16F10, kindly provided by Professor José Daniel Lopes from the Department of the Microbiology and Immunology of the Federal University of São Paulo (UNIFESP), was cultured in RPMI 1640 medium (Gibco, Carlsbad, CA, USA) pH 7.3, supplemented with 10% fetal bovine serum (FBS, Gibco, Carlsbad, CA, USA) at 37 °C in a humidified atmosphere with 5% of CO_2_.

Fish oil was purchased from Naturalis^®^ (Nutrição&Farma LTDA, Sao paulo, Brazil) and, according to the manufacturer, 2 g of fish oil containedf a total of 1.6 of polyunsaturated fatty acids, 1.1 g of eicosapentaenoic acid (EPA), and 0.2 g of docosahexaenoic acid (DHA). More than 80% of fish oil is composed by omega-3 polyunsaturated fatty acids. The soybean oil was purchased from Soya Distribuidora de Produtos Alimentícios LTDA and, according to the manufacturer, 12.3 g of soybean oil contained a total of 6.7 g polyunsaturated fatty acids, 6.0 g of linoleic acid (omega-6), and 0.7 g of linolenic acid (omega-3). More than 89% of soybean oil is composed by omega-6 polyunsaturated fatty acids.

Cell culture was performed by an addition of 100 uL of RPMI 1640 medium, supplemented with 10% fetal bovine serum, containing 5000 B16F10 melanoma cells (5 × 10^3^ cells) per well in a 96-well cell culture plaque. After 24 h, different amounts (40, 20, 10, and 5 uL) of fish oil rich in omega-3 or soybean oil rich in omega-6 or a mixture of these oils in a ratio of 1:1 were added to the B16F10 cell culture. The final volume of each well was standardized to 200 uL and the cell culture was incubated for 48 h. It is important to clarify that the maximum amount of oil (fish or soybean) used here was chosen based on a previous report by de Sá et al. [37]. In this study, the authors supplemented C57BL/6 mice three times a week, with a dose of fish oil at 2 g per kg of weight. The volume of fish oil administrated by oral gavage was around 40 uL.

The cell viability assay was developed following a standard protocol [38]. Melanoma cell viability was determined after incubation with the MTT reagent (0.5 mg/mL, Sigma, St Louis, MO, USA), its solubilization with SDS 1% and absorbance detection.

### 4.2. Animals

C57BL/6 female mice (aged from 6 to 8 weeks) were kindly provided from the University of São Paulo Medical School (FM-USP) and maintained under 12 h daylight cycles without food or water restriction at the Institute of Physical Activity and Sports Science, belonging to the Cruzeiro do Sul University (UNICSUL). It is worthy clarifying that two independent experiments were performed using at least five mice in each group.

### 4.3. Evaluation of the Effect of Omega-3 and Omega-6 on Tumor Progression

Tumorigenicity assays were performed in agreement with that which has been previously described [3]. To assess the effect of omega-3 and omega-6 on tumor progression, mice were coinjected with 1 × 10^6^ of apoptotic B16F10 cells mixed with 1 × 10^3^ of viable B16F10 cells and treated by gavage with 10 uL of water (control group) or fish oil rich in omega-3 or soybean oil rich in omega-6 or a mixture of soybean and fish oil (1:1 ratio).

The first round of treatment by gavage was performed 2 h before the subcutaneous coinjection of melanoma cells in C57BL/6 female mice and repeated on the four consecutive days. The gavage treatment was performed using a number 23 steel gavage tube and a 1.0 mL microsyringe.

Based on Bachi et al. [3], after 14 days of coinjection, the tumor growth was monitored once a week and the mice presenting a subcutaneous mass above 4 mm^3^ were considered positive for the presence of tumors. 

### 4.4. Tumor Homogenate

After 21 days of the coinjection, the tumor was surgically removed, weighted, mixed with 5 mL of phosphate-saline buffer (PBS 1×, pH 7.3), and macerated. The homogenate solution obtained was centrifuged for 4 min at 1400 rpm, and aliquots of 1 mL of the supernatant were collected and stored at −80 °C for further analysis.

### 4.5. Determination of Cytokines and Inflammatory Mediators in the Tumor Homogenate

The concentration of the cytokines IL-1β, IL-6, IL-10, and CXCL1 (R&D System, Minneapolis, MN, USA) and the inflammatory mediators LTB_4_, LTB_5_, PGE_2_, and PGE_3_ (MyBioSource, San Diego, CA, USA) were determined in the tumor homogenate by an ELISA kit. The values of the cytokines were normalized by the total content of proteins, as measured by the Bradford method [39], using the ratio of total protein (in milligrams) to cytokine concentrations in each sample. In a similar way, the values of inflammatory mediators were normalized by the total tumor weight, using the ratio of total weight (in grams) to inflammatory mediator concentrations in each sample.

### 4.6. Statistical Analysis

All data obtained in this study are expressed as the mean and standard deviation (mean ± SD) and were used in the non-paired nonparametric student’s t-test to analyze the statistical difference between the means using GraphPad Prism version 7.0 for Windows (GraphPad, San Diego, CA, USA). Differences at *p*-values < 0.05 were considered as statistically significant.

## Figures and Tables

**Figure 1 ijms-20-03765-f001:**
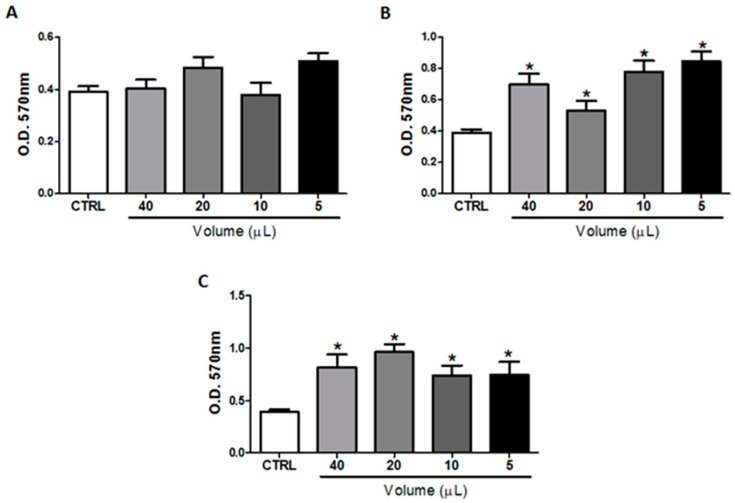
Cell viability assay. B16F10 cells were cultured with different concentrations (40, 20, 10, and 5 μL) of fish oil rich in omega-3 (**A**) or soybean oil rich in omega-6 (**B**) or a mixture of these oils in a ratio of 1:1 (**C**) for 48 h. The risk value α was set at 5% (*p* < 0.05). * *p* < 0.05.

**Figure 2 ijms-20-03765-f002:**
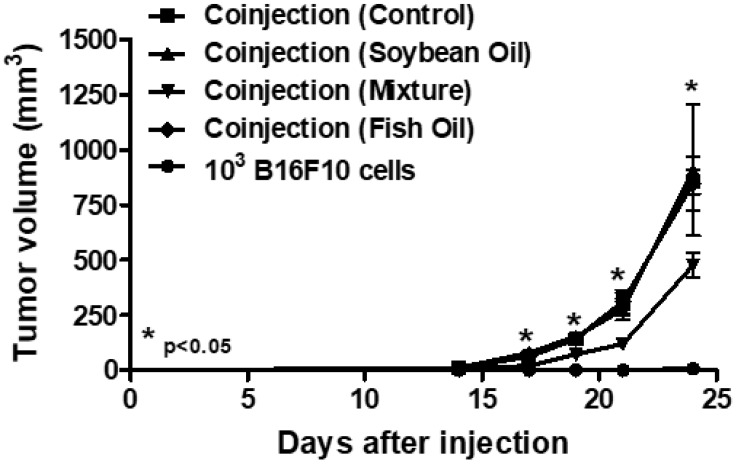
Melanoma growth assay. C57BL/6 mice were coinjected with 10^6^ B16F10 apoptotic cells plus a sub-tumorigenic dose of 10^3^ B16F10 viable melanoma cells in a subcutaneous tissue. After the coinjection, the mice were treated by gavage during five consecutive days with fish oil rich in omega-3 or soybean oil rich in omega-6 or a mixture of these oils in a ratio of 1:1. All experiments were carried out using groups of five 6–8 week old female mice in, at least, two independent experiments, and it was considered as harboring tumors when the subcutaneous mass reached 4 mm^3^ (palpable tumor). The risk value α was set at 5% (*p* < 0.05). * *p* < 0.05.

**Figure 3 ijms-20-03765-f003:**
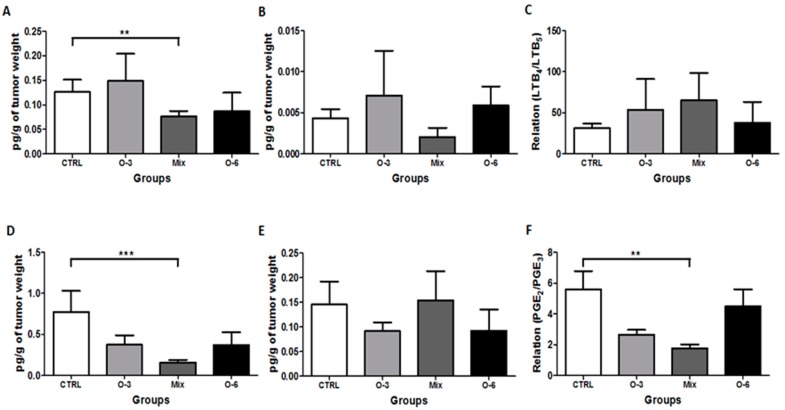
Determination of inflammatory mediators. C57BL/6 mice were coinjected with 10^6^ B16F10 apoptotic cells plus a sub-tumorigenic dose of 10^3^ B16F10 viable melanoma cells in a subcutaneous tissue. After the coinjection, the mice were treated by gavage during five consecutive days with fish oil rich in omega-3 or soybean oil rich in omega-6 or a mixture of these oils in a ratio of 1:1. After 21 days, the tumors were removed and a homogenate was obtained to evaluate the concentration of LTB_4_ (**A**), LTB_5_ (**B**), the ratio between LTB_4_ and LTB_5_ (LTB_4_/LTB_5_) (**C**), prostaglandin E_2_ (PGE_2_) (**D**), prostaglandin E_3_ (PGE_3_) (**E**), and the ratio between PGE_2_ and PGE_3_ (PGE_2_/PGE_3_) (**F**) in the tumor microenvironment. All experiments were carried out using groups of five 6–8 week old female mice in, at least, two independent experiments. The risk value α was set at 5% (*p* < 0.05). ** *p* < 0.01, *** *p* < 0.001.

**Figure 4 ijms-20-03765-f004:**
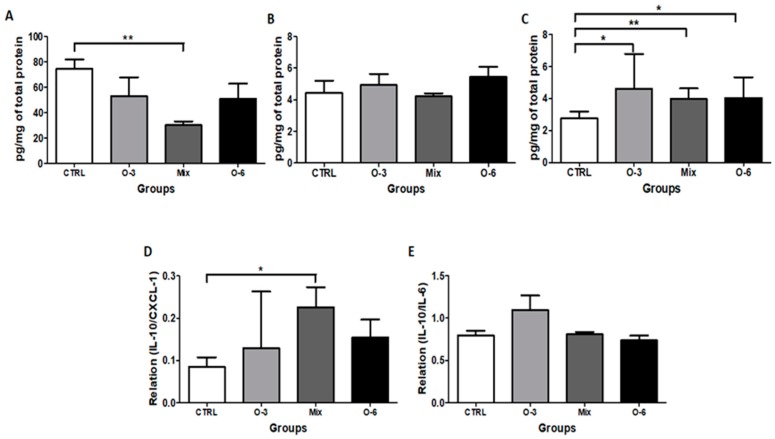
Determination of cytokines in vivo. C57BL/6 mice were coinjected with 10^6^ B16F10 apoptotic cells plus a sub-tumorigenic dose of 10^3^ B16F10 viable melanoma cells in a subcutaneous tissue. After the coinjection, the mice were treated by gavage during five consecutive days with fish oil rich in omega-3 or soybean oil rich in omega-6 or a mixture of these oils in a ratio of 1:1. After 21 days, the tumors were removed and a homogenate was obtained to evaluate the concentration of CXC ligand 1 (CXCL1) (**A**), interleukin 6 (IL-6) (**B**), IL-10 (**C**), the ratio between IL-10 and CXCL1 (IL-10/CXCL1) (**D**) and the ratio between IL-10 and IL-6 (IL-10/IL-6) (**E**) in the tumor microenvironment. All experiments were performed using groups of five 6–8 week old female mice in, at least, two independent experiments. The risk value α was set at 5% (*p* < 0.05). * *p* < 0.05, ** *p* < 0.01.

**Figure 5 ijms-20-03765-f005:**
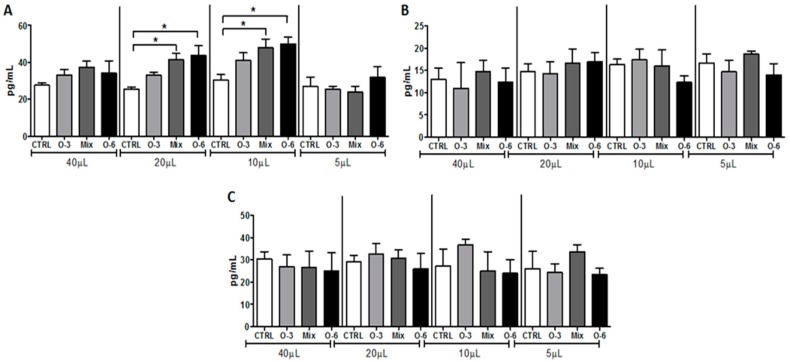
Determination of cytokines in melanoma cell culture. B16F10 cells were cultured with different concentrations (40, 20, 10, and 5 μL) of fish oil rich in omega-3 or soybean oil rich in omega-6 or a mixture of these oils in a ratio of 1:1 for 48 h to determine the concentration of IL-10 (**A**), IL-6 (**B**), and CXCL1 (**C**) levels. The risk value α was set at 5% (*p* < 0.05). * *p* < 0.05.

**Figure 6 ijms-20-03765-f006:**
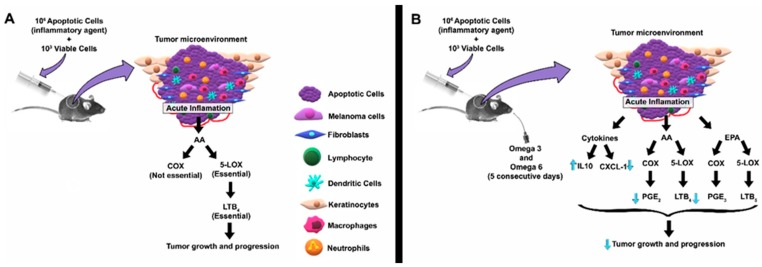
Representative illustration of the main findings of our group. In (**A**), the previous results published by Bachi et al. [3] are presented. In (**B**), the results are shown. Arachidonic acid (AA); –cyclooxygenase (COX); CXC motif chemokine ligand 1 (CXCL1) -; interleukin 10 (IL-10); eicosapentaenoic acid (EPA); –lipoxygenase (LOX); leukotriene B_4_ (LTB_4_); leukotriene B_5_ (LTB_5_); –prostaglandin E_2_ (PGE_2_); prostaglandin E_3_ (PGE_3_).

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
