# Peer review of "A Mixture of Polyunsaturated Fatty Acids ω-3 and ω-6 Reduces Melanoma Growth by Inhibiting Inflammatory Mediators in the Murine Tumor Microenvironment"

_ijms, 2019, doi:10.3390/ijms20153765_

Round 1
Reviewer 1 Report
The results of the in vivo study showed that a mixture of omega-3 and omega-6 reduces melanoma growth are novel and interesting. However, the experimental procedure did not use purified fatty acids but used fish oil for omega-3 and soybean oil for omega-6. These oils contain other fatty acids aside from omega-3 and omega-6 fatty acids. Therefore, the conclusion that a mixture of omega-3 and omega-6 reduces melanoma growth is questionable. The conclusion should be revised in that oral administration of 1:1 mixture of fish oil and soybean oil reduces melanoma growth in mice but fish oil alone or soybean oil alone is ineffective.
The dose of the oils used in the study was 10 ul per mouse by gavage (line 327). How is it possible to administer this volume accurately to a mouse using a syringe and a gavage needle with a dead volume which could be 1 or 10 ul depending upon the gauge of the gavage needle. Part of the 10 ul dose of the oil will just remain in the dead volume of the needle.
The in vitro study using melanoma cells in 96-well cell culture plates is questionable. The oils are virtually insoluble in aqueous media. The culture media contain 10% FBS which may increase the solubility of the oils in the media. The findings of the dose-response study using 5, 10, 20 and 40 ul of oil per well may be meaningless if 5 ul is already saturating when added to 0.2 ml of culture medium/well. The volume of culture medium per well was not stated in the Methods section but 0.2 ml/well is a standard volume used in cell culture using 96-well plates. What is the solubility of the oils in RPMI 1640 medium with 10% FBS?
Author Response
Please see the attachment. All the alterations in the manuscript are marked in yellow.

Reviewer 2 Report
I have read with interest the paper by Almeida and coll, that is focused on the effects of ω-3 and ω-6 against melanoma growth. The experiments have been conducted both in cell cultures as well in murine model (C57BL/6 mice) and the effect on cytokines (IL-6, IL-10 and CXCL-1) and on inflammatory mediators (Leucotrienes and prostaglandins) have also been evaluated.
Unless the research methods are adequate and well discussed, I believe that the effect of fatty acids has not been properly investigated: as it is stated in the Materials and Methods section, the effect of ω-6 alone has not been evaluated. Hence, the experiments were conducted by the use of: i) fish-oil rich in ω-3, ii) soybean oil rich ω-6, but containing both ω-3 and ω-6, and iii) a mixture of fish-oil and soybean oil in 1:1 ratio. On this basis, even when only soybean was administerd in cell culture media or injiected in mice, it is not composed only by ω-6. Since the effect of the mixture reached the best effect against melanoma growth, one can argue that the effect could be due to the ω-6 alone. How can we sustain that the obtained results are due to the mixture or just to the ω-6? This is an important point to clarify, and further experiments should be performed. In case this would not be possible, I believe that clearly explained in the text.
Another important point to be discussed is the fact that in the in vivo experiments the cytokine/ inflammatory mediators have been measured in the tumor homogenate. Why the tumor was not excised and evaluated in totoby immunohistochemistry? This latter could allow to properly mark the presence and the localization of inflammation (mainly cytokine) into the melanoma tissue. Hence, the intratumoral or peritumoral inflammatory infiltrate should also be evaluated.
Moreover, why the authors did not use the B16 mouse (instead of C57BL/6) as experimental model?
Again, since the use of fatty acids to inhibit the melanoma growth should represent a promising future treatment, the Authors might briefly discuss such point in the Discussion. Do they believe that the ω-3 and ω-6 should also be effective when administered as topical ointment?
In my opinion the description of the results depicted in Fig 3A and 3B should be revised (lines 127-128) and the statistical significance should be stressed.
Line 155: please change higher with lower (please check it in the corresponding Fig 4D).
Figure 5: I believe that the Graph is too much full of data, I suggest to show only the measurements obtained with 10 microliter, or maybe those with 10 and 20 μL (please remember that the mice where treated with the concentration of 10 μL). Another option is to report the other data as Supplementary Materials.
Line 206: please correct the sentence “it has was shown effectively suppress"
Author Response

(The authors gave the same response as above.)

Round 2
Reviewer 1 Report
The authors have addressed all my concerns. However, minor text editing is needed for typos and grammar in the new version of the manuscript.
Reviewer 2 Report
The questions have been addressed